# Synthesis of metalla-dual-azulenes with fluoride ion recognition properties

Hai-Cheng Liu[1,3], Kaidong Ruan[1,3], Kexin Ma[1], Jiawei Fei[1], Yu-Mei Lin [1] ✉ & Haiping Xia [1,2] ✉

Azulene-based conjugated systems are of great interests due to their unusual structures and photophysical properties. Incorporation of a transition metal into azulene skeleton presents an intriguing opportunity to combine the $d_\pi$-$p_\pi$ and $p_\pi$-$p_\pi$ conjugated properties. No such metallaazulene skeleton however has been reported to date. Here, we describe our development of an efficient [5 + 2] annulation reaction to rapid construction of a unique metal-containing [5-5-7] scaffold, termed metalla-dual-azulene (MDA), which includes a metal-laazulene and a metal-free organic azulene intertwined by sharing the tropylium motif. The two azulene motifs in MDA exhibit distinct reactivities. The azulene motif readily undergoes nucleophilic addition, leading to N-, O- and S-substituted cycloheptanetrienyl species. Demetalation of the metallaazulene moiety occurs when it reacts with $^n$Bu$_4$NF, which enables highly selective recognition of fluoride anion and a noticeable color change. The practical [5 + 2] annulation methodology, facile functional-group modification, high and selective fluoride detection make this new π-conjugated polycyclic system very suitable for potential applications in photoelectric and sensing materials.

In the more than two centuries since the first aromatic compound, naphthalene was discovered, there has been a continuous interest in the synthesis and research of the properties of various fused-ring skeletons. Azulene is a non-benzenoid hydrocarbon isomer of naphthalene that has attracted significant attention from chemists because of its unique properties and reactivities which arise from the fusion of an electron-rich 5-membered ring and an electron-deficient 7-membered ring, producing a compound with a dipole moment of 1.08 D[1,2]. This structure has been used extensively for the construction of azulene-based π-functional materials, polycyclic aromatic hydrocarbons, and heteroaromatics[3–5]. Cyclopenta[cd]azulene (CPA), a [5-5-7] fused-tricyclic framework that consists of two azulene moieties fused to the same 7-membered ring is of particular interest (Fig. 1a). It was first synthesized by Hafner et al. in 1959[6] and its structure was subsequently confirmed in 1970[7]. Only a few synthetic methods and reactivity studies concerning CPA have since been reported[8–13], but because the tricyclic [5-5-7] skeleton and its derivatives have been

found in fullerenes[14,15], nanographenes[16,17], natural products and drug-like molecules[18,19], designing and exploring the reactivity of the [5-5-7] skeleton is of great significance.

Metal-containing polycyclic π-conjugated systems continue to attract considerable attention[20–23]. The incorporation of transition metal with $d$-orbitals into the organic polycyclic framework results in $d_\pi$–$p_\pi$-conjugated patterns, providing both organic and organometallic characteristics and leading to special physicochemical properties and possible applications[24]. In recent decades, significant progress has been made in the development of polycyclic metallacycles with 5-membered or 6-membered ring motifs[25–28], including metallanaphthalene[29,30], metallaanthracene[31,32], metallapentalenes[33,34], and other motifs[35–37]. However, hydrocarbon seven-membered ring, especially the azulene motif, has not been observed in these polycyclic metallacyclic systems, notwithstanding a few reports of metallacycloheptatrienes[38–40]. Further research could potentially lead to the discovery of new compounds with unique properties and reactivities.

[1]State Key Laboratory of Physical Chemistry of Solid Surfaces, College of Chemistry and Chemical Engineering, Xiamen University, 361005 Xiamen, Fujian, China. [2]Department of Chemistry, Shenzhen Grubbs Institute, Southern University of Science and Technology, 518055 Shenzhen, China. [3]These authors contributed equally: Hai-Cheng Liu, Kaidong Ruan. ✉e-mail: linyum@xmu.edu.cn; hpxia@xmu.edu.cn

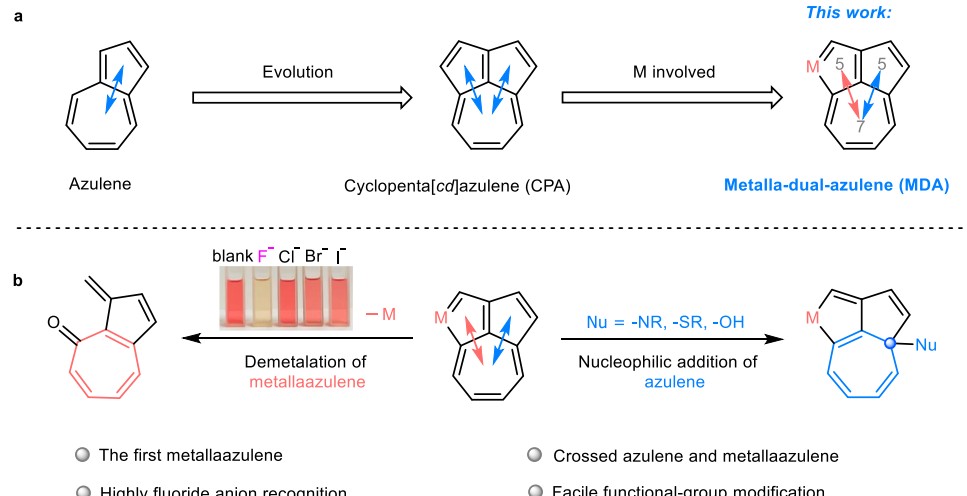

**Fig. 1 | Overview of this work. a** Structural skeletons of azulene, cyclopenta[*cd*]azulene (CPA) and metalla-dual-azulene (MDA). **b** The reactivities of MDA, demetalation of MDA by treatment with ^nBu4NF along with the noticeable color change of the solution (left), and nucleophilic addition of MDA with various reagents (right).

Herein, we present a convenient [5 + 2] annulation strategy which can be used to construct an uncommon metal-containing [5-5-7] skeleton termed a metalla-dual-azulene (MDA). This structure can be viewed as a cross between a metallaazulene and an organic azulene, with both sharing a seven-membered ring (Fig. 1a). The replacement of a carbon by a transition metal in the [5-5-7] skeleton can have a significant impact on the dipole moment of the molecule, and shows that the two azulene fragments exhibit completely different reactivities. The organic azulene motif allows for facile installation of various substituents in the 7-membered ring through nucleophilic addition, providing a wide range of functional group diversity and fine-tuning of the compounds' electronic properties (Fig. 1b, right). Demetalation of the metallaazulene moiety can be achieved by treatment with ^nBu4NF, leading to a distinct and visible color change. This recognition of a fluoride ion is not impeded in any of the twelve different anions, and the limit of detection (LOD) of fluoride has been determined to be $9.73 \times 10^{-7}$ M (18.44 ppb), thereby providing a sensitive method for detection of fluoride ions in solution (Fig. 1b, left).

## Results

### Synthesis of precursor (1) and preparation of metalla-dual-azulene compounds (2a–2h) by [5 + 2] annulation of 1 with alkynes

Treatment of the diyne (**L**) with [Ir(CH3CN)(CO)(PPh3)2]BF4 in CH2Cl2 under N2 at r.t. for 6 h resulted in the formation of compound **1** with an isolated yield of 81% (Fig. 2a). Single-crystal X-ray analysis indicated that **1** contains a tricyclic skeleton of an iridacyclopentadiene fused two rings (Fig. 2c). A possible mechanism for the formation of **1**, involving the [2 + 2 + 1] cycloaddition of **L** with a simple metal complex is proposed (Supplementary Information−p. 9).

Compound **1** contains a bent phenyl allyl alcohol fragment, which serves as a five-carbon synthon with potential reactivity toward unsaturated substrates. Accordingly, treatment of **1** with phenylacetylene and AgBF4 in the presence of HBF4·Et2O at r.t. in CH2Cl2 for 15 min led to the formation of compound **2a** with an isolated yield of 87% (Fig. 2a). The reaction involves a formal [5 + 2] cycloaddition of the phenyl allyl alcohol in **1** with phenylacetylene, constructing a seven-membered ring, and leading to the formation of a π-conjugated fused pentacyclic iridium compound. To investigate the generality of this reaction, various alkynes with different groups were investigated (Fig. 2b). Phenylacetylene with an electron-donating (−OMe) substituent or an electron-deficient group (−CF3) at the *para*-position proved to be

good substrates, giving the corresponding products in good yields (90% for **2b**, 85% for **2c**). 4-Ethynylbiphenyl, naphthylene-2-acetylene, and 3-ethynylthiophene were also tolerated well and afforded the targeted product (**2d**–**2f**) in good yields. The internal alkynes, 1-phenyl-1-propyne, and 1-phenyl-1-pentyne also reacted smoothly, affording the desired products in excellent yield (82% for **2g**, 84% for **2h**). Both aryl-substituted terminal and internal alkynes are well compatible in the reactions, however, when substituting the alkynes with alkyl groups (e.g. 1-heptyne, ethoxyethyne, and 3-butyn-2-one), there are no products observed (Supplementary Information, pp. 37–38).

Single-crystal X-ray diffraction analysis revealed that compounds **2a**–**2d** contain an extended π-conjugate pentacyclic system, comprised of a [5-5-7] tricyclic iridacycle fused with one phenyl ring and attached to a second phenyl ring (Fig. 2c and Supplementary Information, pp. 45–48). The central [5-5-7] skeleton can be regarded as a metallaazulene combined with an organic azulene by sharing the seven-membered ring, resulting in a unique metalla-dual-azulene compound such as **2a**–**2d**. The Ir−C bond lengths in **2a** (Ir1−C1: 2.115(11) Å, Ir1−C10: 2.182(9) Å) are almost identical, while the C−C bond lengths within the two fused five-membered rings (1.351(14)−1.478(13) Å) and the newly formed seven-membered ring (1.392(14)−1.442(14) Å) are intermediate between typical C−C single and C=C double bond lengths[23,35,41], indicating a delocalized [5-5-7] structure. The seven-membered ring is almost planar, as reflected by the mean deviation of 0.086 Å from the least-squares plane. The C−C bond distances and the planarity suggest it is an aromatic tropylium motif[42], indicating a significant contribution from the resonance form **2'**.

The structures of compounds **2a**–**2h** were also confirmed by nuclear magnetic resonance spectroscopy (NMR) and high-resolution mass spectrometry (HRMS) (Supplementary Information, pp. 9–15). The signal from the proton at C6 in **2a** was observed at $\delta = 6.69$ ppm in the aromatic proton region, showing a distinct shift to the lower field when compared to traditional alkenes. In the $^{13}$C{$^1$H} NMR spectrum of **2a**, the signal (227.3 ppm) from C10 was observed at a significantly lower field than that of C1 (176.6 ppm). This is probably because C10 is situated in a positive tropylium environment, which further suggests the resonance contribution of **2'**. The $^{31}$P{$^1$H} NMR spectrum shows only a singlet at −2.6 ppm, attributed to IrPPh3.

### Experimental and theoretical investigations of the mechanism

Control experiments were conducted to elucidate the mechanism of the reaction, including the identification of key intermediates

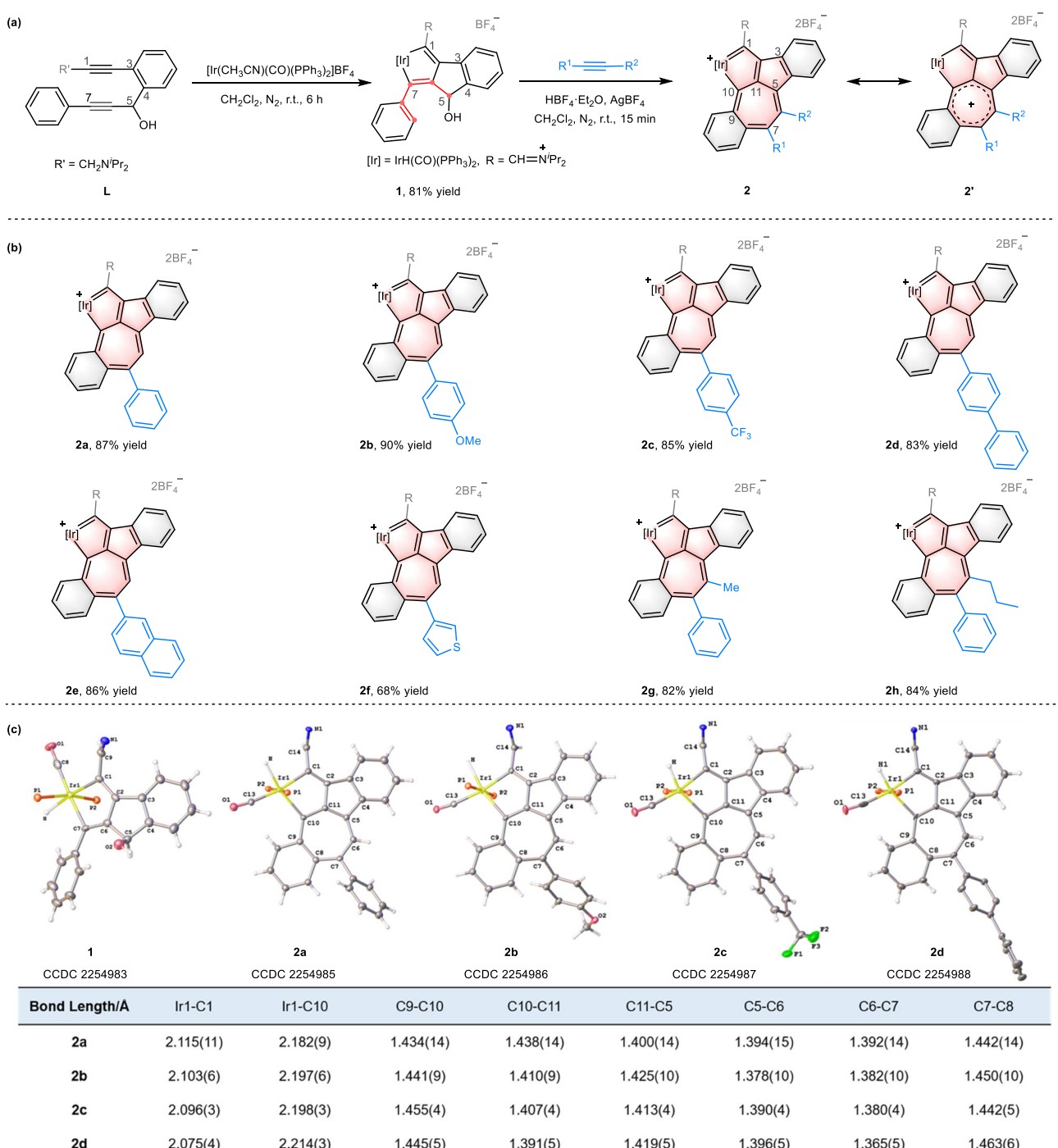

**Fig. 2 | Synthesis and characterization of precursor 1 and compounds 2a–2h.**
**a** and **b** Synthesis of compound **1** and preparation of **2a–2h** by reactions of **1** with different alkynes. **c** X-ray structures for the cations of compounds **1, 2a–2d** (thermal ellipsoids drawn with 50% probability, the phenyl groups in PPh$_3$ and isopropyl groups in −CH=N$^+$($^i$Pr)$_2$ were omitted for clarity) and the selected bond lengths (Å) for **2a–2d**. Selected bond distances (Å) for **1**: Ir1–C1 2.183(4), Ir1–C7 2.139(4), C1–C2 1.362(6), C2–C6 1.447(5), C6–C7 1.362(6), C2–C3 1.468(6), C3–C4 1.398(6), C4–C5 1.519(6), C5–C6 1.530(6).

and the detection of hydrogen. Stepwise addition of HBF$_4$·Et$_2$O and AgBF$_4$ was performed to examine their respective roles in the reaction. First, treatment of **1** and phenylacetylene with HBF$_4$·Et$_2$O at r.t. in CH$_2$Cl$_2$ resulted in a 93% yield of a new compound (**3a**) (Fig. 3a). The X-ray crystal structure of **3a** shows that it has a pentacyclic skeleton similar to that of **2a**, except that the seven-membered ring is a cycloheptatrienyl motif. This is confirmed by the typical C–C single bond lengths of C11–C5 (1.509(6) Å) and C5–C6 (1.501(7) Å) (Fig. 3b). The signal attributed to the hydrogen at

C5 was observed at $\delta$ = 3.28 ppm in the $^1$H NMR spectrum. Subsequently, treatment of the isolated **3a** with AgBF$_4$ in CH$_2$Cl$_2$ led to the corresponding compound (**2a**) in 89% yield. Hydrogen produced in the in situ reaction was detected by gas chromatography (GC) (Fig. 3b right), and the formation of silver mirrors was observed after the reaction (Supplementary Information, p. 16). These results suggest the reaction initially proceeds through [5 + 2] cycloaddition promoted by HBF$_4$·Et$_2$O to form a cycloheptatrienyl intermediate, which then undergoes oxidative dehydrogenation enabled by

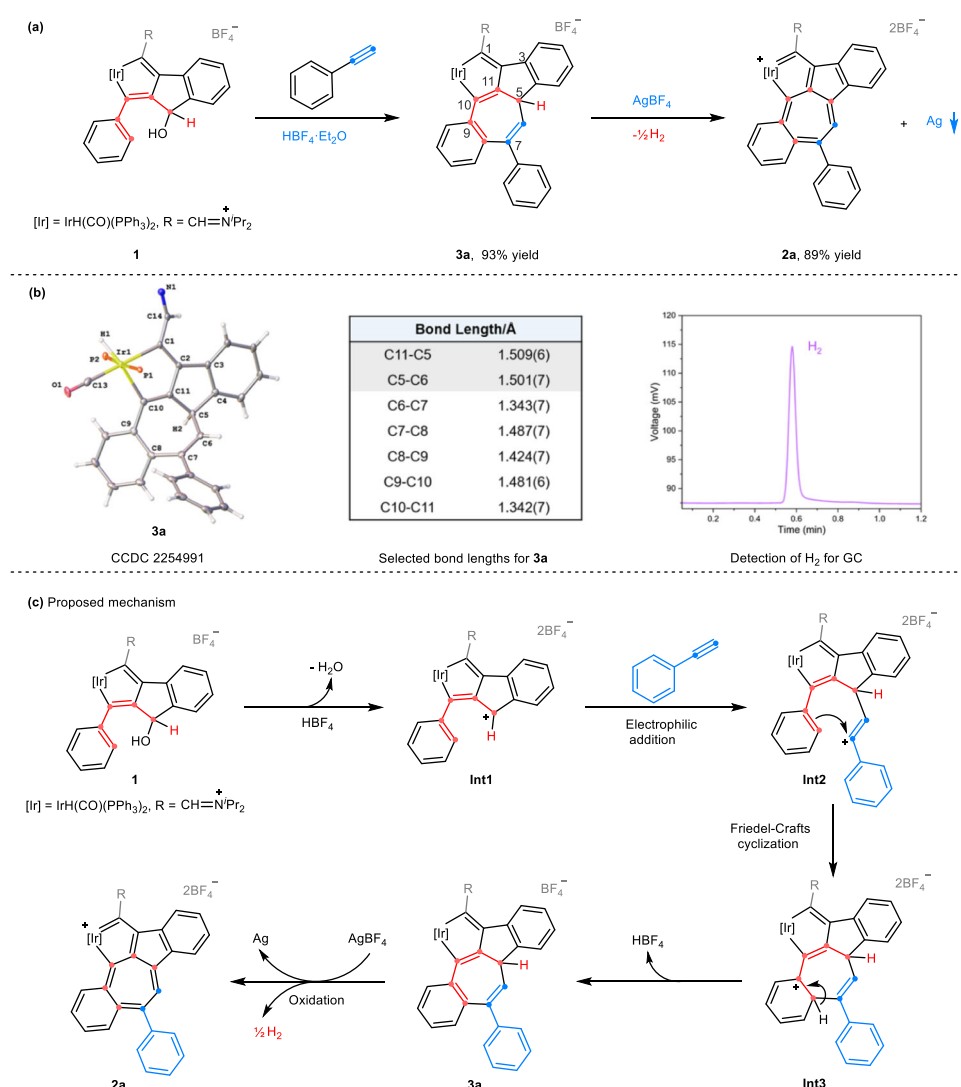

**Fig. 3 | Control experiments and proposed mechanism. a** Isolation of intermediate **3a** and its further conversion to **2a** by stepwise addition of HBF₄·Et₂O and AgBF₄ in the reaction. **b** Structure of the cation of **3a** (thermal ellipsoids drawn with 50% probability, the phenyl groups in PPh₃ and isopropyl groups in −CH = N⁺(ⁱPr)₂

omitted for clarity) (left), selected bond lengths (Å) for **3a** (middle), detection of H₂ by GC (gas chromatography) in the conversation of intermediate **3a** to **2a** (right). **c** Proposed mechanism for the formation of **2a**.

AgBF₄. The latter process can also proceed in the presence of other oxidants, such as MnO₂ or AgOTf.

Based on these experimental results, a plausible mechanism was proposed for the formation of **2a** (Fig. 3c). The process begins with the dissociation of the hydroxyl group in **1** by HBF₄·Et₂O, giving **Int1**. Next, the electrophilic addition of **Int1** to phenylacetylene affords vinyl cation **Int2**, which is a common intermediate observed in alkynes participating reactions[43–45]. Subsequently, intramolecular electrophilic attack within **Int2** generates a [5 + 2] annulated cation species **Int3**. The loss of a proton from the cation completes the Friedel−Crafts cyclization, resulting in the formation of stable **3a**. Finally, oxidative hydride abstraction of **3a** by AgBF₄ affords the target product (**2a**). Further evidence for the proposed mechanism involving electrophilic addition via **Int2** has been provided and directly calculating the [5 + 2] cyclization pathway has been ruled out (Supplementary Information, pp. 37–38). Development of efficient and straightforward methods for constructing seven-membered rings has practical significance. Previously reported methods for construction of 7-membered rings via [5 + 2] cycloaddition have utilized vinylcyclopropanes[46–48], 3-acyloxy-1,4-enynes[49,50] and metal η⁵-pentadienyl compounds[51] as the five-carbon synthons. However, methods to synthesize tropylium ions are

relatively limited and often require multiple reactions[52–54]. In this case, [5 + 2] cycloaddition via a Friedel−Crafts cyclization and aromatization provides a new and efficient route with which to build the tropylium motif. The use of an iridacyclopentadiene derivative (**1**) as the five-carbon synthon enables access to a new metal-incorporated [5-7] bicyclic skeleton, which is an unprecedented metallaazulene motif.

DFT calculations were conducted to gain a deeper understanding of the mechanism. The computed Gibbs free energy profile of the essential reaction steps is shown in Fig. 4. In the HBF₄·Et₂O promoted reaction, the conversion of reactant **1** to **Int3** is energetically favored and achieved smoothly with no high barriers lowering the potential energies. Two possible pathways can lead from **Int3** to product **2a**. The direct formation of **2a** through **TS3**' has a high barrier of 19.3 kcal mol⁻¹ (Fig. 4, gray line). In contrast, the formation of intermediate **3a** through **TS3** is kinetically favored, with a negligible barrier of 0.1 kcal mol⁻¹, and results in the formation of stable compound (**3a**) with an exothermicity of 18.2 kcal mol⁻¹ (Fig. 4, black line). The oxidative dehydrogenation of **3a** by AgBF₄ affords the 7-membered ring in the final product **2a**. The aromaticity of this 7-membered motif has been evaluated by calculated results of the nucleus-independent chemical shift (NICS)

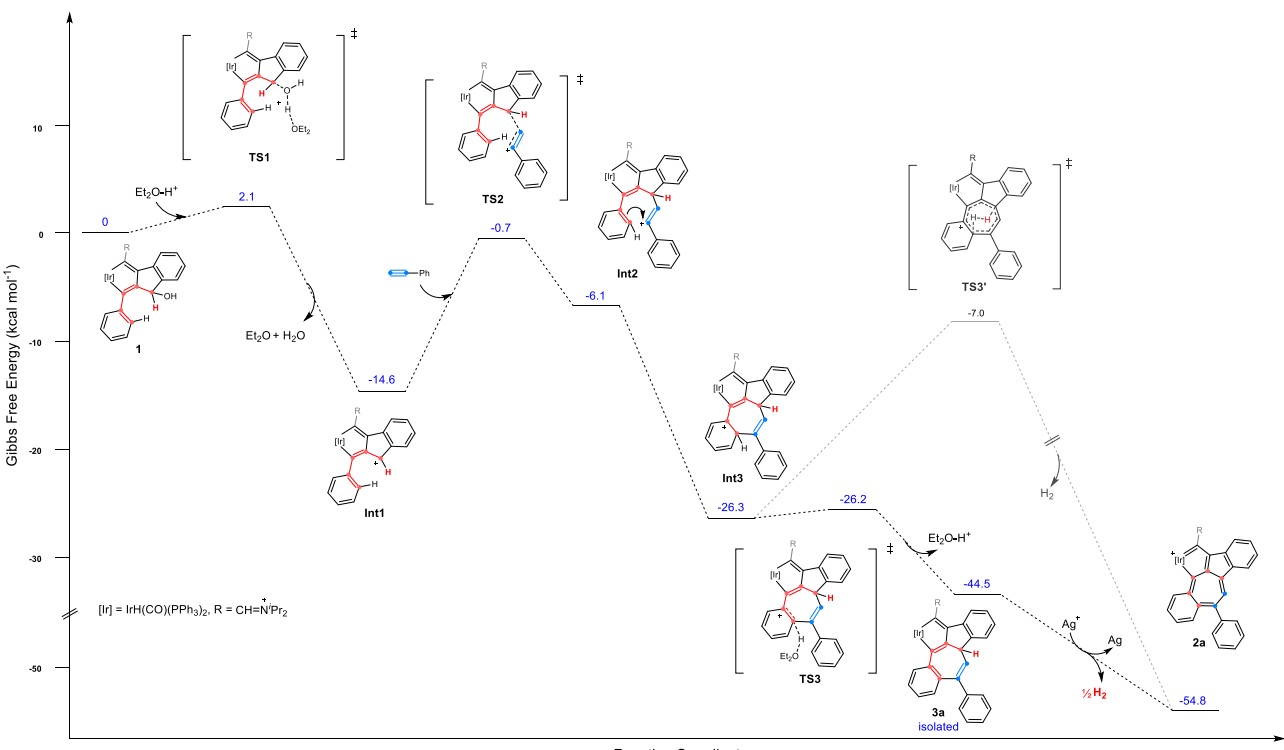

**Fig. 4 | DFT calculations for mechanistic investigation.** Gibbs free energy profile for the formation of compounds **3a** and **2a**. Energies are given in kcal mol⁻¹. The Gibbs free energy of reactant **1** with HBF₄·Et₂O was designated as 0 kcal mol⁻¹. The B3LYP-D3BJ/Def2-TZVP method with the SMD solvation method in CH₂Cl₂ was used.

values and the anisotropy of the current-induced density (ACID) (Supplementary Information, p. 35).

### Nucleophilic addition of metalla-dual-azulene with versatile –N, –S, and –O reagents

The presence of a cationic tropylium moiety in metalla-dual-azulene makes it susceptible to nucleophilic reactions. Indeed, treatment of **2a** with NaOH in CH₂Cl₂ for 15 min resulted in the formation of a product (**4a**) in high yield (96%) (Fig. 5a). The X-ray crystal structure of **4a** revealed a pentacyclic skeleton with a hydroxyl group attached at C5. The bond lengths of C5–C11 (1.513(6) Å) and C5–C6 (1.494(6) Å) indicate the typical C–C single bonds (Fig. 5b and Supplementary Information, p. 49). The 7-membered ring is extremely torsional, as reflected by the large mean deviation of the least-squares plane (0.303 Å), in contrast to the high planarity of the precursor **2a** (0.086 Å). Nucleophilic addition to the tropylium species resulted in the formation of a cycloheptatrienyl motif. Notably, compound **4a** can be efficiently transferred back to **2a** by treating with HBF₄·Et₂O (Supplementary Information, p. 27). The result indicates that the reversible conversion between compounds **2a** and **4a** can be achieved by regulating the acidity and basicity of the system. Other nucleophiles, including aniline, sodium thiomethoxide, and sodium thiophenoxide were also reacted with **2a**, and the desired products (**5a, 6a**, and **6b**) were obtained in 95%, 90%, and 81% yields, respectively. A condensed dual descriptor (CDD) study showed that both C5 and C7 sites are electron-deficient (Supplementary Information, p. 35). We performed further calculations to evaluate the main contributions of resonance structures (**2a-I–2a-IV**) with positive charge distribution on the seven-membered ring (7MR) using the natural resonance theory analysis method (NRT). The results revealed that resonance structure **2a-I** with the positive charge on C5 had the highest contribution, accounting for a maximum proportion of 22.53% compared to other structures (**2a-I–2a-IV**), which explains the

nucleophilic addition occurred exclusively at the C5 site (Supplementary Information, p. 36).

A direct and convenient route to form **4a** was developed, starting from **1** which was treated with phenylacetylene and HBF₄·Et₂O with AgBF₄ at r.t. in CH₂Cl₂ for 15 min. The mixture was filtered and concentrated, and then subjected to chromatography on a basic alumina column, giving compound **4a** with an isolated yield of 85%. An ¹⁸O labeling experiment showed that the hydroxyl group (–¹⁸OH) attached to C5 of compound **4** originated from the H₂O in the system, which was confirmed by HRMS (Supplementary Information, pp. 25–26). Different alkynes with various functional groups were investigated (Fig. 5c). Phenylacetylenes bearing electron-donating substituents (–Me, –tBu, or –OMe) at *para*-positions were shown to be excellent substrates, giving the corresponding products (**4b, 4c**, and **4d**) in yields of 89%, 90%, and 91%, respectively. Electron-withdrawing groups (–CF₃) or halogen (–F) and (–Ph) substituents at *para*-positions or *meta*-positions also demonstrated good applicability as substrates with yields of 89%, 85%, and 87% for **4e, 4f**, and **4g**, respectively. The polycyclic 1-naphthylethyne and heterocyclic 2-ethynylthiophene reacted smoothly and afforded the desired products **4h** and **4i** in 89% and 61% yields, respectively. The internal alkyne (1-phenyl-1-propyne) was also compatible and gave the corresponding product (**4j**) with an 86% yield. The structures of all the compounds (**4a–4j**) were confirmed by NMR spectroscopy, HRMS, and the structures of **4a, 4d, 4f**, and **4j** were further confirmed by X-ray diffraction (Supplementary Information, pp. 49–52).

### UV–Vis absorption spectra of products and fluoride anion recognition properties of 2a

The UV–Vis absorption spectra showed that the pentacyclic compounds all exhibit strong absorption bands. As depicted in Fig. 6a, the characteristic energy absorption band of **2a** in the low-energy absorption regions (537 nm) is red-shifted by ~39 nm compared to the precursor **1** (498 nm) due to the extended π-conjugated

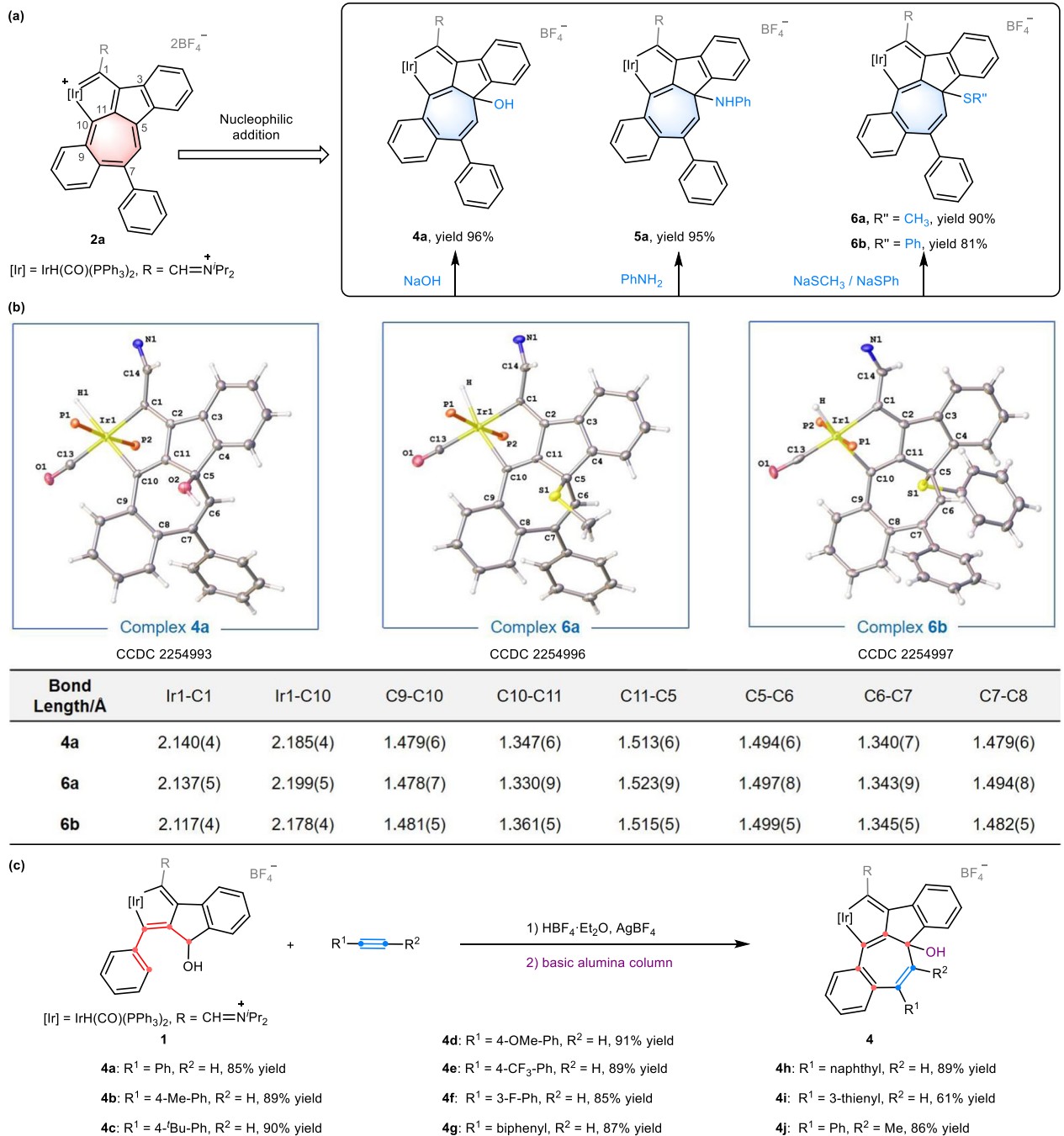

**Fig. 5 | The reactivities study of 2a and the direct synthesis of compounds 4a–4j. a** Nucleophilic addition of **2a** with versatile nucleophiles (NaOH, PhNH₂, and NaSR"). **b** X-ray structures for **4a**, **6a**, and **6b** (drawn with 50% probability level) and the selected bond lengths (Å). **c** Direct synthesis of –OH substituted products **4a–4j** by reaction of **1** with various alkynes.

framework in **2a**. The molar absorption coefficient of **2a** at 537 nm is $1.4 \times 10^4 \, M^{-1} \, cm^{-1}$. The characteristic energy absorption could be fine-tuned by modifying the substituents on the seven-membered ring. Specifically, the electron-donating group (–OMe) in $R^1$ of **2b** resulted in a significantly red-shifted absorption at 565 nm when compared with compound **2c**, which contains an electron-withdrawing group (–CF₃) (528 nm). The absorption spectra of dearomatized counterparts (**4a**, **5a**, **6a**, and **6b**) exhibited similar absorption properties (Fig. 6a right), but the molar absorption coefficient value ($\varepsilon$) of characteristic energy absorption varied from 1.4 to $1.9 \times 10^4 \, M^{-1} \, cm^{-1}$.

To further analyze the absorption spectra, we implemented time-dependent DFT calculations at the B3LYP-D3BJ/Def2-TZVP level for compounds **2a** and **4a** (Fig. 6b, c). The calculated excited wavelength $\lambda_{max} = 515 \, nm$ of **2a** was found to be 95.8% attributable to HOMO → LUMO transitions, with a corresponding oscillator strength ($f$) of $f = 0.2561$. Meanwhile, the calculated excited wavelength $\lambda_{max} = 509 \, nm$ of **4a** was found to be 97.4% attributable to HOMO → LUMO transitions, with a corresponding oscillator strength of $f = 0.2414$. Notably, the nucleophilic addition to the tropylium motif in **2a** led to the formation of a cycloheptatrienyl motif with a $C(sp^3)$ site in **4a**. However, despite this change, the electron delocalization of **4a** remains similar to that of **2a** over the polycyclic skeletons. Additionally, their HOMO–LUMO gap is nearly identical ($\Delta E = 2.88 \, eV$ for **2a**, $\Delta E = 2.89 \, eV$ for **4a**). This explains the similarity of the absorption maxima of **2a** and **4a** in the low-energy

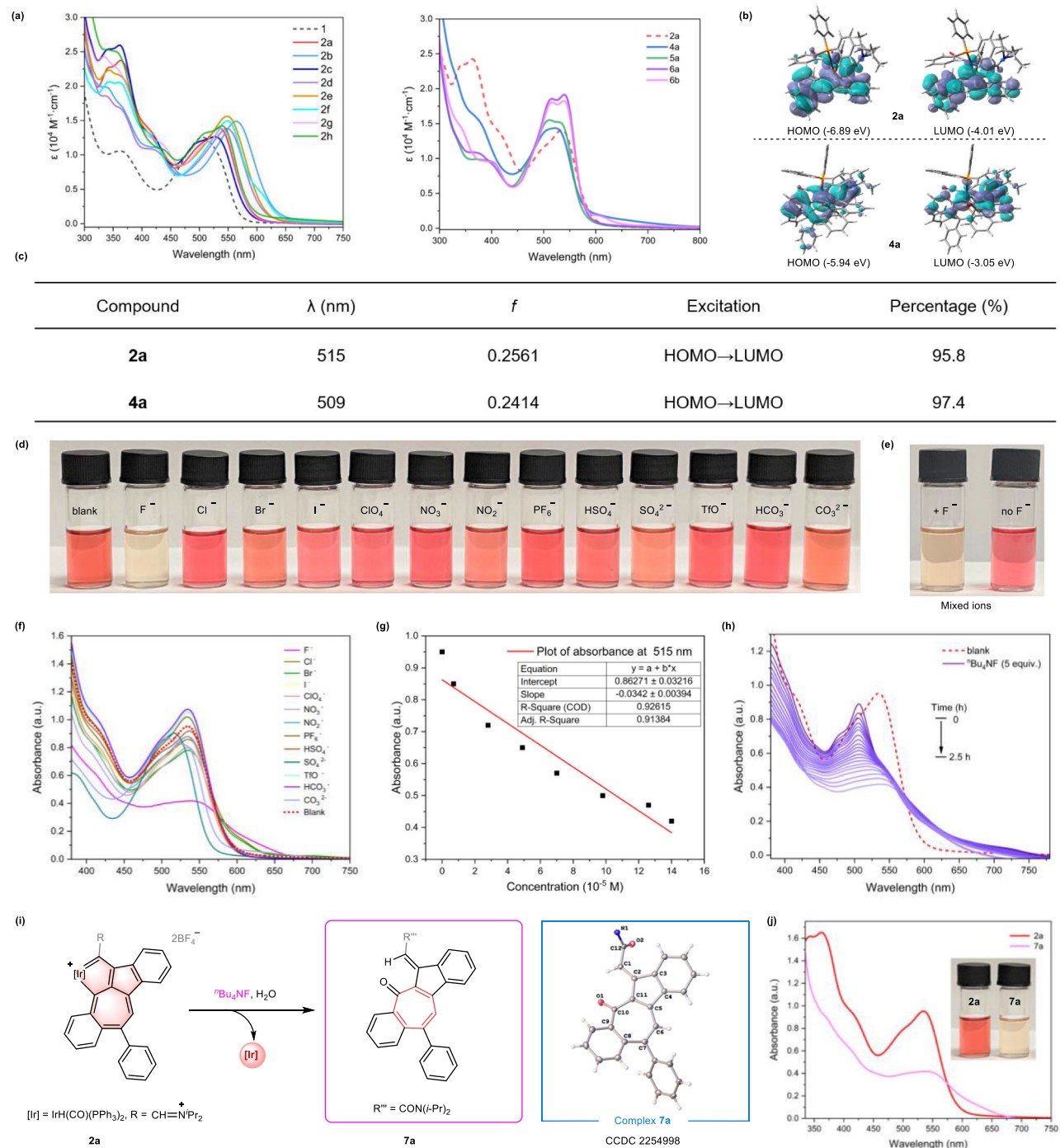

**Fig. 6 | Optical properties and fluoride anion recognition. a** UV–Vis absorption spectra of compounds **1, 2a–2h, 4a, 5a, 6a**, and **6b** measured in CH₂Cl₂ at r.t. (1.0 × 10⁻⁵ M). **b** Selected highest occupied molecular orbitals (HOMO) and lowest unoccupied molecular orbitals (LUMO) related to the excitation of compound **2a** and **4a** (isovalue = 0.02) are given. The eigenvalues of the MOs (molecular orbitals) are given in parentheses. **c** Excited wavelength (λ) oscillator strengths (*f*) and related wavefunctions. **d** Color changes of **2a** (0.07 mM) by the addition of 5.0 equiv. of various anions (0.35 mM) including F⁻, Cl⁻, Br⁻, I⁻, ClO₄⁻, NO₃⁻, NO₂⁻, PF₆⁻, HSO₄⁻, SO₄²⁻, TfO⁻, HCO₃⁻ and CO₃²⁻ of ⁿBu₄NX salts in CH₂Cl₂ for 5 min,

respectively. **e** Mixing the solutions with each of the above 12 anions (left) and without the addition of F⁻ (right). **f** UV–Vis absorption spectra of **2a** (0.07 mM) after the addition of 5.0 equiv. of different anions in CH₂Cl₂. **g** Linear relationship for absorbance at 515 nm for **2a** versus the concentration of ⁿBu₄NF (7.0–35 μM). **h** The in situ UV–Vis absorption of **2a** (0.07 mM) and ⁿBu₄NF (5.0 equiv.) in CH₂Cl₂ was monitored within 2.5 h. **i** Isolation of product **7a** by reaction of **2a** with ⁿBu₄NF. X-ray structures for **7a** (drawn with 50% probability level). **j** UV–Vis absorption spectra of **2a** and **7a** in CH₂Cl₂ and the corresponding photographs of solutions (inset).

absorption regions. The theoretical spectra are thus consistent with the trends observed in the experimental spectra. The electrochemical properties of these pentacyclic compounds have also been investigated. The complexes exhibit redox processes, however, are found to be irreversible (Supplementary Information, p. 40).

Compounds **2a–2h** in the solid state are all stable below 130 °C in air for at least 4 h (Supplementary Information, p. 41). They have a cationic tropylium motif and strong absorptions in the visible range, which led us to explore their potential as colorimetric chemosensors through interactions with anions. Upon adding 5.0 equiv. of tetra-*n*-

butylammonium salts (${}^{n}$Bu$_4$NX), including X = Cl$^-$, Br$^-$, I$^-$, ClO$_4^-$, NO$_3^-$, NO$_2^-$, PF$_6^-$, HSO$_4^-$, SO$_4^{2-}$, TfO$^-$, HCO$_3^-$, and CO$_3^{2-}$ to a solution of **2a**, weak or negligible color changes were observed for these anions. However, when mixed with fluoride anion (F$^-$), a rapid and obvious change to essentially colorless occurred within 5 min (Fig. 6d). The UV−Vis absorption spectra showed a significant decrease in the absorption bands around 515 nm in the presence of fluoride anion (Fig. 6f). The solutions displayed no noticeable color change when **2a** was added to a mixture of 12 anions without F$^-$ (Fig. 6e, right), but the color faded immediately upon addition of F$^-$ to the anionic mixture (Fig. 6e, left).

These results indicate that **2a** has promising selective recognition of F$^-$ in an organic solvent, and is not affected by other anions. Based on the results from the absorbance titration, it was concluded that the detection limitation of **2a** for fluoride anion was sensitive. A linear curve was fitted showing the decreased absorbance versus the concentration of F$^-$ ranging from 7.0 to 35 μM under the same sensing conditions (Fig. 6g). The limits of detection (LOD) were calculated as $3\sigma/k$, where $k$ is the slope of the linear fitting and $\sigma$ is the standard deviation of blank measurements. It was found that **2a** had a detection limit of $9.73 \times 10^{-7}$ M (18.44 ppb) for F$^-$, implying a highly sensitive detection ability (Fig. 6g and Supplementary Information, pp. 32−33). The in-situ UV−Vis spectroscopy showed that the characteristic absorption peak at ~515 nm decreased gradually over 2.5 h when 5.0 equiv. F$^-$ was added (Fig. 6h). These results suggest that **2a** can detect F$^-$ through color changes that are easily observed with the naked eye or by colorimetric measurement. As fluoride ions are commonly present in aqueous solutions, we conducted further investigations to evaluate the effect of water on the recognition of fluoride ions. By adding different amounts of water (0.01−2.0 mL) into the reaction systems containing **2a** and ${}^{n}$Bu$_4$NF, the color of the solution changed rapidly from red to light yellow, which was similar to the observation in the blank reaction where no water was added. The UV−Vis absorption spectra of the resulting solutions also exhibit no significant changes compared to the blank sample (Supplementary Information, p. 33). The findings indicated that the fluoride ion recognition capacity off.

To investigate the nature of the specificity of **2a** towards F$^-$, we conducted experiments to isolate the product of the reaction of **2a** with ${}^{n}$Bu$_4$NF and succeeded in obtaining a product (**7a**). The X-ray crystal structure of **7a** revealed that it is an organic tetracyclic framework without the iridium fragment (Fig. 6i). An ${}^{18}$O labeling experiment showed that the oxygen atom of the carbonyl group (C=O) attached to C10 in compound **7a** originates from the H$_2$O present in the system (Supplementary Information, p. 31). In the presence of ${}^{n}$Bu$_4$NF, demetalation of **2a** may be triggered by the interaction of the fluoride anion with the metal center, subsequently inducing the demetalation process (Supplementary Information, p. 32). Fluoride ion is considered less nucleophilic in comparison to −O, −N, and −S nucleophilic reagents (Fig. 5a), therefore, fluoride ion does not proceed through direct nucleophilic addition in this case. Notably, retention of **2a** for other ${}^{n}$Bu$_4$NX salts (X = Cl, Br, I) as evidenced by there being essentially no change in the UV−Vis spectra. Consistently, previous research has shown that the tropylium moiety is capable of detecting fluoride ions out of its halide family[55]. The development of fluoride probes is of considerable significance in health and environmental issues[56,57], and previously reported F$^-$ sensors have been designed based on the interactions between fluoride anions and Lewis acids, fluoride−hydrogen bonding[58], fluoride−silicon bond interaction, and other types of materials[59,60]. In the present case, a new reaction-based colorimetric fluoride sensor involving demetalation has been developed, leading to color responses that allow for detection by the naked eye (Fig. 6j). The metal interacting with a conjugated system enables a significant enhancement of the absorption and color change, which is a prerequisite for detection.

## Discussion

We have developed an efficient [5 + 2] annulation reaction to rapidly build a unique metal-containing [5-5-7] scaffold, known as metalla-dual-azulene. This is composed of a metallaazulene and an organic azulene that share the tropylium motif. The reaction involves a Friedel−Crafts cyclization and oxidative dehydrogenation under mild conditions. The resulting fused-polycyclic metallacycles exhibit strong absorption bands in the UV−Vis spectra, with characteristic energy absorption bands that can be finely tuned by modifying the substituents at C5, C6, and C7 on the seven-membered ring. In a mixture of 13 available anions, the metalla-dual-azulene (**2a**) demonstrates highly selective recognition of the fluoride anion with noticeable color changes that allow for detection by the naked eye. The findings of this study offer a valuable strategy for the construction of seven-membered rings and create new opportunities for enlarging the π-conjugated polycyclic systems with fascinating properties.

## Methods
### General information
Details of the synthesis and characterization of the multiyne (**L**) and metallapolycyclic compounds (**1, 2a-2h, 3a, 4, 5a, 6a, 6b**, and **7a**) can be found in the supplementary information, pp. 5−31. For X-ray data of all the described compounds (**L, 1, 2a-2d, 3a, 3x, 4a, 4d, 4f, 4j, 6a, 6b**, and **7a**) see supplementary information, pages 42-67. For HRMS, ${}^{1}$H, ${}^{31}$P{${}^{1}$H} NMR, and ${}^{13}$C{${}^{1}$H} NMR spectra of compounds in this article, see Supplementary Information, pp. 69−117 and Supplementary Figs. 37−134. For the thermal decomposition data of compounds **2a-2h, 3a, 4a, 5a, 6a**, and **6b**, see Supplementary Information, p. 41.

### General procedure for preparation of compound 2
Alkyne (0.24 mmol) and HBF$_4$·Et$_2$O (0.10 mmol) were added to a mixture of **1** (235.4 mg, 0.20 mmol) and AgBF$_4$ (116.4 mg, 0.60 mmol) in CH$_2$Cl$_2$ (2.0 mL), which was stirred at r.t. under nitrogen for 15 min to give a brown solution. The solution was filtered through Celite, and the filtrate was concentrated under a vacuum. The resulting residue was purified by column chromatography (silica gel (200−300 mesh), eluent: dichloromethane/acetone = 2:1) to afford compounds **2**.

### General preparation procedure of compounds 4a, 5a, 6a, 6b
A mixture of **2a** (269.4 mg, 0.20 mmol) and 1.2 equiv. nucleophilic reagents (NaOH, PhNH$_2$, NaSCH$_3$, NaSPh, respectively) in CH$_2$Cl$_2$ (2.0 mL) were stirred at r.t. under nitrogen for 2−6 h to give a pink solution. Then the solution was filtered by Celite, and the filtrate was concentrated under a vacuum. The resulting residue was purified by reverse precipitation performed with hexane or column chromatography (alumina (200−300 mesh)) to afford the target product. Yield: 96% for **4a**, 95% for **5a**, 90% for **6a**, 81% for **6b**, respectively).

## Data availability
The authors declare that the data supporting the findings of this study are available within this article and its Supplementary Information file. The X-ray crystallographic coordinates for structures reported in this study have been deposited at the Cambridge Crystallographic Data Centre (CCDC), under deposition numbers 2254984 (**L**), 2254983 (**1**), 2254985 (**2a**), 2254986 (**2b**), 2254987 (**2c**), 2254988 (**2d**), 2254991 (**3a**), 2254992 (**3x**), 2254993 (**4a**), 2254994 (**4d**), 2254995 (**4f**), 2262455 (**4j**), 2254996 (**6a**), 2254997 (**6b**), 2254998 (**7a**). These data can be obtained free of charge from the Cambridge Crystallographic Data Centre via www.ccdc.cam.ac.uk/data_request/cif. Extra data are available from the corresponding author upon request. Source data of cartesian coordinates, UV−Vis absorption, cyclic voltammetry, and gas chromatography are provided in this paper. Source data are provided with this paper.

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

## Acknowledgements

This work is supported by the Natural Science Foundation of China (Nos. 22071206, 92156021, and 21931002), the Natural Science Foundation of Fujian Province of China (No. 2020J01025), the Shenzhen Science and Technology Innovation Committee (no. JCYJ20200109140812302), Guangdong Provincial Key Laboratory of Catalysis (no. 2020B121201002), Introduction of Major Talent Projects in Guangdong Province (2019CX01C079) and the Financial Support for Outstanding Talents Training Fund in Shenzhen.

## Author contributions

H.X. and Y.-M.L. designed and conceived the project. H.-C.L. and K.M. performed the experiments. H.-C.L., J.F., Y.-M.L., and H.X. analyzed and interpreted the experimental data. K.R. performed the theoretical calculations. All of the authors discussed the results and contributed to the preparation of the manuscript.

## Competing interests

The authors declare no competing interests.
