## [Peer Review File · Nature Communications]

Synthesis of Metalla-Dual-Azulenes with Fluoride Ion Recognition PropertiesReviewers' Comments:

Reviewer #1:

Remarks to the Author:

In this study, the authors developed a synthesis of metallaazulene derivatives via [5+2] cyclization and investigated their reactivity. The metallaazulenes prepared undergo de-metallization upon reaction with $n\text{Bu}_4\text{NF}$ and exhibit a pronounced color change due to the formation of a tropone skeleton. These facts mean that metallaazulenes are highly selective in recognizing fluoride anions and is suitable for application in sensing materials.

Based on a comprehensive evaluation of the above points, the reviewer recommends that this paper be published in Nature Communications after some minor revisions.

Revision points are shown in below.

1. Fig 3 (c): The authors insist that Int2 is formed by electrophilic addition to Int1. However, Int2 is a vinyl cation and appears to be thermodynamically unstable. Do the authors have any evidence to support that this reaction is via Int2?

2. Fig 4: Related to the previous question, the pathway from Int2 to TS2 seems to have a relatively high activation energy. Furthermore, the process from Int1 to Int2 (vinyl cation) is also endothermic and seems to be energetically unfavorable. Can the authors provide a reasonable explanation for these processes?

3. Fig 5 (a) and Figure S7: Can the authors explain why the C5 position in 2a is more electron deficient? As a side note, the structure of 2a in Fig. 5 is different from that of 2a shown in Fig. 2 (b). The current form may cause misreading by the readers and should be unified with one or the other structure.

4. Fig 6 (f) and (i): Why does the nucleophilic attack of the fluoride anion proceed at C1 and not at C5 as shown in Fig 5(a)? Further fluoride anion addition is carried out in CH_2Cl_2 solvent, but tropone product 7a is formed. What is the carbonyl oxygen in 7a derived from?

5. Have you examined the voltammetry experiments for the series of compounds prepared in this study? Since some of the azulene derivatives exhibit unique redox properties, it would be worthwhile to perform voltammetry experiments.

Reviewer #2:

Remarks to the Author:

This work by Xia et al reports a new metal-containing [5-5-7] scaffold comprising of a metallaazulene and a metal-free azulene that share the tropylium motif. This n -conjugated polycyclic structure has not been previously reported, and was achieved using a convenient [5+2] annulation strategy. This approach for building a 7-membered ring is significant and the formation mechanism has been thoroughly analyzed through a combination of experimental and computational studies. Additionally, the two azulene fragments exhibited distinct reactivities. Impressively, the metallaazulene moiety permitted highly selective recognition of fluoride anion. All these features highlighted the novelty of this new azulene-based architecture and the potential application. I believe that the manuscript would be well suited for Nature Communications with the following minor revisions:

(1) Compound 2a can undergo nucleophilic addition in the presence of a base, resulting in the formation of compound 4a with a hydroxyl group at C5 site. The question is whether compound 4a can lose the hydroxyl group under acidic conditions, allowing for the reversible conversion between 2a and 4a through acid-base regulation.

(2) Based on the DFT calculations of the mechanism, it appears that the energy barrier for the conversion of Int3 to 2a through TS3' is relatively low. In the absence of AgBF₄ and with the addition of HBF₄, there is a likelihood of generating 2a. Can heating facilitate and promote this conversion?

(3) Compound 2a exhibits excellent performance in recognizing fluoride ions. As fluoride ions are commonly present in aqueous solutions, it is important to consider whether the presence of water affects this specific recognition capability.

(4) The reaction for the formation of 2a demonstrates substrate generality, primarily focusing on various phenyl alkynes. However, it remains uncertain whether substituting the alkyne with an alkyl group would be feasible within this context.

Reviewer #3:

Remarks to the Author:

The manuscript by Lin and Xia describes the fluoride colorimetric sensing properties of a novel family of iridium-dual-azulene.

I believe that the results are extremely interesting, and presented in an impeccable way. I would therefore recommend the publication of the paper in NatComms. However, the authors describe in detail the sensing properties of one of the compounds but they do not mention what happens with the others and which is (if there is) a correlation between the substituent on the central core and the sensing activity/selectivity.

Response Letter

REVIEWER COMMENTS

Reviewer #1 (Remarks to the Author):

In this study, the authors developed a synthesis of metallaazulene derivatives via [5+2] cyclization and investigated their reactivity. The metallaazulenes prepared undergo de-metallization upon reaction with nBu₄NF and exhibit a pronounced color change due to the formation of a tropone skeleton. These facts mean that metallaazulenes are highly selective in recognizing fluoride anions and is suitable for application in sensing materials.

Based on a comprehensive evaluation of the above points, the reviewer recommends that this paper be published in Nature Communications after some minor revisions.

Revision points are shown in below.

1. Fig 3 (c): The authors insist that Int2 is formed by electrophilic addition to Int1. However, Int2 is a vinyl cation and appears to be thermodynamically unstable. Do the authors have any evidence to support that this reaction is via Int2?

Response: Thanks for your kind advice. We have followed the reviewer's suggestion and conducted further investigations into the mechanism. Based on the following considerations, we propose that the reaction proceeds electrophilic addition via **Int2**. Firstly, the presence of a phenyl group attached to **Int2** is essential for the stabilization of the vinyl cation. Our calculation showed that propyne with a methyl group was used to interact with **Int 1**, the energy of the system gradually increased as the distance between the C5 and C6 atoms shortened, indicating that the formation of chemical bond between these atoms is not possible (Figure S17). This align well with experimental observations where no products were obtained when alkynes were substituted with alkyl groups (e.g., 1-heptyne, ethoxyethyne and 3-butyne-2-one). The stronger stabilizing effect of the phenyl group on the vinyl cation (**Int2**), attributed to π - π conjugation, compared to an alkyl group accounts for this discrepancy. Secondly, our calculations indicate that **Int2** is thermodynamically unstable, which can generate **Int3** with no barrier after only passing the simple conformational change as shown in Figure S18 (a \rightarrow b). Consequently, **Int2** cannot be experimentally captured. Thirdly, previous literatures (e.g. references 43-45) have shown that vinyl cations are commonly observed intermediates in carbocation-alkyne reactions (page 7, lines 142-143). The proposed mechanism involving electrophilic addition through **Int2**, supported by calculations, experimental observations and literature references, provides a reasonable explanation for the reaction pathway. We have included the new findings in the manuscript (page 7, lines 146-154) and Supplementary Information (SI, pages S37-38).

Figure S17. Energy diagram of the interaction between propyne and **Int1**.

Figure S18. Conformational change of the phenyl group from **Int2** to **Int3**.

2. Fig 4: Related to the previous question, the pathway from **Int2** to **TS2** seems to have a relatively high activation energy. Furthermore, the process from **Int1** to **Int2** (vinyl cation) is also endothermic and seems to be energetically unfavorable. Can the authors provide a reasonable explanation for these processes?

Response: Thank the reviewer for this observation. As the explanations shown in the response for the question 1, the reaction pathway was supported by calculations, experimental observations and literature references.

We also attempted to calculate the mechanism using a direct [5+2] cyclization method from **Int1** to **Int3**. However, no matter the initial guess for the transition state or if the virtual frequency of the vibration resembled the desired [5+2] cyclization, we consistently obtained **TS2**. This finding supports the idea that **Int2**, despite being energetically higher in energy than **Int1**, which can generate **Int3** with no barrier after only passing the simple conformational change as shown in Figure S18 (a → b). This conformational change prevents the re-decomposition of **Int2** back into the raw materials (**Int1**) and makes it challenging to capture **Int2** experimentally. Thus, we have

ruled out directly calculating the [5+2] cyclization pathway and further support the proposed reaction pathway involving the formation of **Int2** followed by the simple conformational change to yield **Int3** (pages 7-8, lines 146-154, and SI, pages S37-38).

3. Fig 5 (a) and Figure S7: Can the authors explain why the C5 position in **2a** is more electron deficient? As a side note, the structure of **2a** in Fig. 5 is different from that of **2a** shown in Fig. 2 (b). The current form may cause misreading by the readers and should be unified with one or the other structure.

Response: We appreciate the constructive suggestion from the reviewer. As mentioned in the original manuscript, a condensed dual descriptor (CDD) study showed that both C5 and C7 sites in compound **2a** are electron-deficient (SI, page S35). Furthermore, we performed additional calculations to evaluate the main contributions of resonance structures (**2a-I** ~ **2a-IV**) with positive charge distribution on the seven-membered ring (7MR) using natural resonance theory analysis method (NRT) (Figure S16). The results revealed that resonance structures **2a-I** (with a positive charge at C5) and **2a-II** (with a positive charge at C7) made more significant contributions compared to other resonance structures. This is due to that it could keep aromatic nature of the attached phenyl group in these two forms. The observations are consistent with the findings of the CDD studies. In addition, the resonance structure **2a-I** had the highest contribution, accounting for a maximum proportion of 22.53%. This is because the positive charge on C5 is stabilized through p- π conjugation with one benzene ring and two vinyl groups. Similarly, in resonance structure **2a-II**, the positive charge on C7 can be stabilized through p- π conjugation with a benzene ring and a vinyl group. However, it should be noted that the single crystal structure of **2a** demonstrated the presence of a large torsion angle between the planes of the benzene ring (p) and the 7MR, which hinders effective p- π conjugation. As a result, the contribution of resonance structure **2a-II** is 14.46%. Based on these analyses, we conclude that the C5 position of compound **2a** exhibits greatest electron deficiency. We have updated the revised manuscript accordingly incorporating the related description (Pages 10-11, lines 195-206, SI, page S36) and followed the reviewer's suggestion that have unified with one structure for **2a** (Fig 5 and Fig 6i).

Figure S16. The main contributions of resonance structures (**2a-I** ~ **2a-IV**) with positive charge distribution on the seven-membered ring (7MR) evaluated by natural resonance theory analysis method.

4. Fig 6 (f) and (i): Why does the nucleophilic attack of the fluoride anion proceed at C1 and not at C5 as shown in Fig 5(a)? Further fluoride anion addition is carried out in CH₂Cl₂ solvent, but tropone product **7a** is formed What is the carbonyl oxygen in **7a** derived from?

Response: Thank the reviewer for this observation. As described in the original manuscript, the C5 site of compound **2a** can be attacked by nucleophiles. It is noted that in comparison to -O, -N and -S nucleophilic reagents (Fig 5a), fluoride ion is considered less nucleophilic. Furthermore, the introduction of metal in compound **2a** results in a more dispersed positive charge distribution on the tropylium unit in resonance structure **2a'** compared with conventional organic tropylium ions, thereby reducing its binding ability with nucleophilic reagents. Based on these observations, it is hypothesized that fluoride ion does not proceed through direct nucleophilic addition in this case. Instead, the natural resonance theory analysis suggests that resonance structure **2a**, which exhibits significant positive charge distribution on the metal center, may facilitate interaction with fluoride ion at the metal center, subsequently inducing the demetallation process. Notably, transition-metal fluorides are known to be intermediates with high reactivity. Therefore, the hypothesis proposed is that fluoride ions preferentially interact with the metal center, leading to demetalation, rather than undergoing direct nucleophilic reaction (Page 16, lines 301-308 and SI, pages S31-32).

To investigate the source of the carbonyl oxygen in tropone within compound **7a**, a control isotope label experiment was performed (Figure S9 and S10). The reaction involved treating compound **2a** with anhydrous tetrahydrofuran solution of ${}^n\text{Bu}_4\text{NF}$ in the presence of H_2^{18}O . The ESI-MS spectrum of the resulting product showed molecular ions at $m/z = 486.2430$ for ${}^{18}\text{O}$ -**7a** (calculated value $[\text{C}_{32}\text{H}_{29}\text{N}^{18}\text{O}_2+\text{Na}^+]^+$ at $m/z = 486.2404$). The result confirms that the oxygen atom of carbonyl group (C=O) attached to C10 in compound **7a** originates from the H_2O present in the system. The new results have been added in the revised manuscript (page 16, lines 299-301, SI, page S31)

Figure S9. The reaction of compound **2a** with ${}^n\text{Bu}_4\text{NF}$ in the presence of H_2^{18}O .

Figure S10. Positive-ion ESI-MS spectrum for complex $[^{18}\text{O-7a}]^+$ measured in methanol.

5. Have you examined the voltammetry experiments for the series of compounds prepared in this study? Since some of the azulene derivatives exhibit unique redox properties, it would be worthwhile to perform voltammetry experiments.

Response: Thanks for your kind advice. We followed the reviewer's suggestion and conducted investigations into the electrochemical properties of the pentacyclic compounds (page 14, lines 261-263). The obtained results demonstrate the presence of redox processes in all complexes (Figure S20), however, these processes are found to be irreversible. The influence of phenyl substituents on the reduction potentials is evident. In the case of compound **2b**, where electron-donating $-\text{OCH}_3$ group was introduced, an anodic shift in the reduction potential was observed compared to **2a**. Conversely, the incorporation of an electron-withdrawing $-\text{CF}_3$ group in compound **2c** led to a cathodic shift in reduction potential. Other derivatives such as **4c**, **4f**, and **6b** also exhibit slightly shifts in their reduction potentials compared to that of **2a** (SI, page 39).

Figure S20. Cyclic voltammograms of **2a-2c**, **2h**, **4c**, **4f**, and **6b** in CH_2Cl_2 obtained by the cyclic voltammetry (CV) with a glassy carbon as the working electrode, a platinum rod as the auxiliary electrode, Ag/AgCl as the reference electrode, $[\text{nBu}_4\text{N}]\text{BF}_4$ as the supporting electrolyte and a ferrocene/ferrocenium couple as the external standard.

Reviewer #2 (Remarks to the Author):

This work by Xia et al reports a new metal-containing [5-5-7] scaffold comprising of a metallazulene and a metal-free azulene that share the tropylium motif. This π -conjugated polycyclic structure has not been previously reported, and was achieved using a convenient [5+2] annulation strategy. This approach for building a 7-membered ring is significant and the formation mechanism has been thorough analyzed through a combination of experimental and computational studies. Additionally, the two azulene fragments exhibited distinct reactivities. Impressively, the metallazulene moiety permitted highly selective recognition of fluoride anion. All these features highlighted the novelty of this new azulene-based architecture and the potential application. I believe that the manuscript would be well suited for Nature Communications with the following minor revisions:

(1) Compound **2a** can undergo nucleophilic addition in the presence of a base, resulting in the formation of compound **4a** with a hydroxyl group at C5 site. The question is whether compound **4a** can lose the hydroxyl group under acidic conditions, allowing for the reversible conversion between **2a** and **4a** through acid-base regulation.

Response: Thanks for your constructive advice. In the original manuscript, it was shown that compound **2a** undergoes nucleophilic addition in the presence of a base, leading to the formation of compound **4a** with a hydroxyl group at C5 site. To explore the reversibility of this reaction, we

performed the reaction of **4a** with acid (Figure S7 and S8). The *in situ* $^{31}\text{P}\{^1\text{H}\}$ NMR spectra revealed that by increasing the amount of $\text{HBF}_4\cdot\text{Et}_2\text{O}$ (from 0.5 equiv. to 2.0 equiv.) in the CH_2Cl_2 solution of compound **4a**, the quantity of **4a** gradually decreased while that of **2a** increased. When the acid was added to 2.0 equiv., a complete transformation of **4a** into **2a** was observed. The result indicates that the reversible conversion between compounds **2a** and **4a** can be achieved by regulating the acidity and basicity of the system. We have incorporated these new findings into the revised manuscript (Page 10, lines 189-192, SI, page S27).

Figure S7. The reversible conversion between compounds **2a** and **4a** by controlling the acid and base.

Figure S8. The *in situ* $^{31}\text{P}\{^1\text{H}\}$ NMR spectra of addition varied amounts of $\text{HBF}_4\cdot\text{Et}_2\text{O}$ (from 0.5 equiv. to 2.0 equiv.) in the CH_2Cl_2 solution of compound **4a**.

(2) Based on the DFT calculations of the mechanism, it appears that the energy barrier for the conversion of **Int3** to **2a** through **TS3'** is relatively low. In the absence of AgBF_4 and with the addition of HBF_4 , there is a likelihood of generating **2a**. Can heating facilitate and promote this conversion?

Response: Thank the reviewer for this constructive suggestion. We acknowledge the reviewer's opinion that the energy barrier for the conversion of **Int3** to **2a** through **TS3'** ($19.3 \text{ kcal mol}^{-1}$) appears to be overcome by heating. However, control experiments involving the reaction of compound **1** with phenylacetylene in the presence of $\text{HBF}_4\cdot\text{Et}_2\text{O}$, heated to 80°C for 24 h, even to

7 days, remain led to main product **3a**, like the results obtained at room temperature (Figure S3 and S4). We consider the competitive process from species **Int3** to **3a** through **TS3** exhibiting an extremely low energy barrier (0.1 kcal mol⁻¹), consequently, this process is predominantly governed by kinetic factors rather than thermodynamic considerations. We have incorporated these descriptions into the revised supplementary Information accordingly (SI, pages S16-17).

Figure S3. The reaction of **1** with phenylacetylene in the absence of AgBF₄.

Figure S4. The *in situ* ³¹P{¹H} NMR in the reaction of **1** (23.5 mg, 0.02 mmol) with phenylacetylene (2.6 μL, 1.2 eq.) in the presence of HBF₄·Et₂O (3.0 eq.).

(3) Compound **2a** exhibits excellent performance in recognizing fluoride ions. As fluoride ions are commonly present in aqueous solutions, it is important to consider whether the presence of water affects this specific recognition capability.

Response: We appreciate the reviewer's valuable suggestion and conducted further investigations to evaluate the effect of water on the recognition of fluoride ions. Figure S12 illustrates the results obtained by adding different amounts of water (0.01- 2.0 mL) into the reaction systems containing **2a** and ⁿBu₄NF. The color of the solution changed rapidly from the red to a light yellow, which was similar to the observation in the blank reaction where no water was added. The UV-Vis absorption spectra of the resulting solutions also exhibit no significant changes compared to the blank sample. Based on these findings, it can be concluded that the fluoride ion recognition capacity of compound **2a** is unaffected by the presence of water. We have included a detailed description of these results in the revised manuscript (page 15, lines 286-293 and SI, page 33).

Figure S12 UV-Vis absorption spectra of **2a** (0.07 mM) with 5.0 equiv. $t\text{Bu}_4\text{NF}$ in CH_3CN in the presence of different amounts of water and the color changes of solutions (insert).

(4) The reaction for the formation of **2a** demonstrates substrate generality, primarily focusing on various phenyl alkynes. However, it remains uncertain whether substituting the alkyne with an alkyl group would be feasible within this context.

Response: Thanks for pointing out this observation. We agree that the examples provided in the article mainly focus on aryl-substituted terminal and internal alkynes. Endeavors has been made when substituting the alkynes with alkyl groups (1-heptyne, ethoxyethyne and 3-butyne-2-one), however, there are no products observed. Our calculation showed that propyne with a methyl group was used to interact with **Int 1**, the energy of the system gradually increased as the distance between the C5 and C6 atoms shortened, indicating that the formation of chemical bond between these atoms is not possible (Figure S17). The presence of a phenyl group attached to **Int2** is essential for the stabilization of the vinyl cation. The stronger stabilizing effect of the phenyl group on the vinyl cation (**Int2**), attributed to π - π conjugation, compared to an alkyl group accounts for this discrepancy. The corresponding description have been added in manuscript (page 5, lines 93-95) and Supplementary Information (SI, page S37).

Figure S17 Energy diagram of the interaction between propyne and **Int1**.

Reviewer #3 (Remarks to the Author):

The manuscript by Lin and Xia describes the fluoride colorimetric sensing properties of a novel family of iridium-dual-azulene.

I believe that the results are extremely interesting, and presented in an impeccable way. I would therefore recommend the publication of the paper in NatComms. However, the authors describe in detail the sensing properties of one of the compounds but they do not mention what happens with the others and which is (if there is) a correlation between the substituent on the central core and the sensing activity/selectivity.

Response: Thank the reviewer for this constructive suggestion. We investigated the interaction between compounds **2** bearing different substituents and fluoride ions. As shown in Figure S13, the color of the solutions (**2a-2h**) rapidly changed from red to pale yellow after adding $n\text{Bu}_4\text{NF}$ for 5 minutes. Despite the variations in the substituents, all the compounds demonstrated good recognition capability towards fluoride anions. The UV-Vis absorption spectra analysis revealed that the absorption around 515 nm of **2g** ($R^1 = \text{Ph}$, $R^2 = \text{Me}$) and **2h** ($R^1 = \text{Ph}$, $R^2 = n\text{-propyl}$) displayed a bit more decay compared with that of compound **2a** ($R^1 = \text{Ph}$, $R^2 = \text{H}$) when interacting with the fluoride ion. The studies suggest that different substituents on the phenyl group had no significant effects on the sensing activity of compounds **2**. The new results have been added in the manuscript and Supplementary Information (page 16, lines 293-295 and SI, page S34).

Figure S13 UV-Vis absorption spectra of **2a-2h** (0.07 mM) after the addition of 5.0 equiv. $t\text{Bu}_4\text{NF}$ in CH_2Cl_2 and the color changes of solution (insert).

Reviewers' Comments:

Reviewer #1:

Remarks to the Author:

This is a revised manuscript of a previously submitted one by the authors. I believe that the revised manuscript contains appropriate corrections and additions to the comments and questions raised by the reviewers, including myself. Therefore, I would like to recommend the publication of the revised manuscript because it can be judged that the quality of the revised manuscript is up to the quality required by Nature Communications.

Reviewer #2:

Remarks to the Author:

This is a revised manuscript, in which the authors have properly addressed my concerns and comments for the previous version. Therefore I support publication of this manuscript as it is.

A point-by-point response to issues raised by the referees is summarized below.

REVIEWERS' COMMENTS:

Reviewer #1 (Remarks to the Author):

This is a revised manuscript of a previously submitted one by the authors. I believe that the revised manuscript contains appropriate corrections and additions to the comments and questions raised by the reviewers, including myself. Therefore, I would like to recommend the publication of the revised manuscript because it can be judged that the quality of the revised manuscript is up to the quality required by Nature Communications.

Response: Thank the reviewer for the positive comments.

Reviewer #2 (Remarks to the Author):

This is a revised manuscript, in which the authors have properly addressed my concerns and comments for the previous version. Therefore I support publication of this manuscript as it is.

Response: Thank the reviewer for the positive comments.

We have taken into consideration all of the reviewers' comments. Finally, we would like to extend our thanks again to all of the reviewers for their helpful suggestions and comments.

With kind regards and best wishes,

Haiping Xia